# Impact of Visual Cues on Consumers’ Freshness Perception of Prepared Vegetables

**DOI:** 10.3390/foods13203342

**Published:** 2024-10-21

**Authors:** Xuan Tran, Nicolas Antille, Marine Devezeaux de Lavergne, Cyril Moccand, David Labbe

**Affiliations:** Société des Produits Nestlé S.A., 1000 Lausanne, Switzerland; mxuan.tran99@gmail.com (X.T.); nicolas.antille@rdls.nestle.com (N.A.); marine.devezeauxdelavergne@rd.nestle.com (M.D.d.L.); cyril.moccand@rdls.nestle.com (C.M.)

**Keywords:** vegetable, freshness, consumer

## Abstract

Freshness is an important quality attribute for vegetables. Identifying the sensory drivers for freshness is important to promote vegetable consumption. Due to the lack of research on freshness of prepared vegetables, this study focused on the role of visual cues of common vegetables (carrot, beetroot, bell pepper) on perception of freshness. Twenty-seven vegetables were prepared by varying five factors and photographed in a plate: (1) shape (stick, large cube, small cube), (2) vegetable presence for each of the three vegetables (yes, no), (3) number of vegetables conjointly present in the plate (1, 2, 3), (4) color (green bell pepper, yellow bell pepper), (5) combined vegetables prepared with same or different shapes. Freshness was rated online by 156 consumers. Visual cues leading to the main increase in freshness were the stick shape vs. large and small cubes, the absence of beetroot, and the presence of green bell pepper vs. yellow bell pepper. Overall, it seems that visual cues associated with minimally processed vegetables, such as stick shapes, which allow to recognize the vegetables in comparison to cube shapes, promote freshness. These results are particularly valuable for culinary and catering professionals and food industries involved in the preparation and/or manufacturing of prepared vegetables.

## 1. Introduction

According to the World Health Organization guidelines, at least 400 g per day of fruits and vegetables is suggested for a healthy lifestyle [1] and as a component of a weight-loss diet [2]. However, increasing consumption of vegetables is compromised as often the least favorite food categories due to poor sensory experiences compared with other food groups [3,4].

Among food-related factors, perceived freshness was rated just after taste for its contribution to the purchase interest of consumers [5] and is a key driver of fruit and vegetable acceptance [6]. Enhancing freshness perception may then increase vegetable consumption to reach the recommended amount. Freshness properties were described as a product quality that is usually associated with descriptive words such as “newly made” and “recently prepared or harvested” [7]. In another study, food freshness was defined as the level of closeness of a food product to its original state in terms of distance, time, and processing [8]. So far, research on perceived freshness in vegetables and fruits is mainly focused on specific species freshly picked up, such as apples [9,10,11], strawberries [12], carrots [12,13], wild rockets, and spinach [14], and freshness perception is the consequence of the absence of processing cues by any means.

There is still evidence showing that minimally processed vegetables, such as cut and steamed vegetables, can be perceived as fresh as long as consumers can easily recognize the vegetables [14]. Opportunities may exist to encourage intake of vegetables through mode of preparation enhancing freshness perception, which is the objective of the current research, i.e., increasing freshness perception of minimally processed vegetables. Freshness perception can be modulated by sensory modalities [9,12,15] and was positively associated to appearance and texture attributes but not to flavor except for the presence of putrid off-flavor [14]. The present study focused on visual cues associated with perceptions of freshness in prepared vegetables. Indeed, visual cues generate expectations, so they play a crucial role in predicting vegetable quality and are positively associated with the increase in vegetable consumption [16].

In previous research, the authors investigated the effect of shape manipulation on intake, liking, and purchase intention, but the link with perceived freshness has not been evaluated [17]. As perceived freshness is a driver of vegetable liking [6], bitter vegetables poorly liked, such as beetroot [18,19], might be perceived lower in freshness compared with preferred vegetables through liking and freshness cognitive association. Food and color diversity in a meal is associated to some extent with higher acceptance [20,21] and increased vegetable consumption [16]. The relationship between the number of vegetables on a plate and perceived freshness has never been addressed. Finally, the impact of color on freshness perception for vegetables was evaluated in a third part comparing green bell pepper and yellow bell pepper.

This study innovates in testing for the first time the impact of vegetable shape, type of vegetable, vegetable combination, shape heterogeneity, and vegetable color on perceived freshness. Such modes of preparation are representative of culinary practices applied to vegetables at home, in restaurants, or sold in cans or bags. The related hypotheses below were tested through an online questionnaire presenting pictures of vegetables on a plate prepared in different conditions. H1a–c were based on a main experimental design (Part 1) and hypotheses H2 (Part 2) and H3 (Part 3) based on two additional experimental designs:

**H1a.** 
*Cube shape reduces perceived freshness of vegetables because of its association with commercialized frozen and canned vegetables.*


**H1b.** 
*Perceived freshness of beetroot is lower than for bell pepper and carrot, due to lower liking.*


**H1c.** 
*Vegetable combination increases acceptance and consequently perceived freshness through cognitive association.*


**H2.** 
*Shape heterogeneity enhances perceived freshness as irregularity is perceived as less processed.*


**H3.** 
*The green color is perceived as fresher than other colors, as green is strongly associated with naturality and consequently perceived freshness.*


Results of the tested hypotheses are described in result Section 3.1 (H1a–c), Section 3.2 (H2), and Section 3.3 (H3) and then discussed in relation to the aim of generating knowledge dedicated to restaurant chefs, prepared vegetable manufacturers, and healthcare professionals. The ultimate goal is to enhance the perceived freshness of prepared vegetables and, consequently, encourage greater consumer intake for a healthier diet.

## 2. Materials and Methods

### 2.1. Raw Materials (Vegetables)

Three common vegetables were selected for this study: carrot, beetroot, and bell pepper. These vegetables were chosen based on the ability to prepare them in similar shapes (see Table 1) and their color varieties. The company Léguriviera (Montreux, Switzerland) provided the fresh-cut vegetables that were stored in the fridge at 5 °C before use. Three different types of shape (Table 1) were chosen, as shown below.

### 2.2. Experimental Design

Separate but overlapping designs of experiments were used to define a relevant set of samples to validate the different hypotheses formulated previously.

Hypotheses H1a, H1b, and H1c: A full factorial experimental design (Table 2) was created using “Shape” (3 levels: stick, large cube, and small cube) and the “Presence/Absence” of each vegetable (2 levels: yes and no) as experimental factors. The initial design contained 24 trials, but the 3 conditions in which all vegetables were absent were discarded for obvious reasons. From these three factors, we derived a fourth factor, calculated as the number of vegetables present on the plate.

**Table 2 foods-13-03342-t002:** Main experimental design for shape (stick, large cube, small cube) and vegetable presence (yes, no). The number of vegetables is derived from the three vegetable absence/presence variables (1, 2, 3).

Trial	Shape	Carrot	Green Bell Pepper	Beetroot	Number of Vegetables
1	Stick	Yes	Yes	Yes	3
2	Stick	Yes	Yes	No	2
3	Stick	Yes	No	Yes	2
4	Stick	Yes	No	No	1
5	Stick	No	Yes	Yes	2
6	Stick	No	Yes	No	1
7	Stick	No	No	Yes	1
8	Large cube	Yes	Yes	Yes	3
9	Large cube	Yes	Yes	No	2
10	Large cube	Yes	No	Yes	2
11	Large cube	Yes	No	No	1
12	Large cube	No	Yes	Yes	2
13	Large cube	No	Yes	No	1
14	Large cube	No	No	Yes	1
15	Small cube	Yes	Yes	Yes	3
16	Small cube	Yes	Yes	No	2
17	Small cube	Yes	No	Yes	2
18	Small cube	Yes	No	No	1
19	Small cube	No	Yes	Yes	2
20	Small cube	No	Yes	No	1
21	Small cube	No	No	Yes	1

Hypothesis H2: Two additional trials were produced to allow comparisons with some of the samples from the core design (Table 2). The result is a second experimental design (Table 3) that allows measuring the impact of shape heterogeneity through direct comparison between samples.

**Table 3 foods-13-03342-t003:** Experimental design for shape diversity evaluation. Two additional samples in white cell were produced: sample 22 for comparison with samples 2 and 16 in gray cell and sample 23 for comparison with samples 1, 8, and 15 in gray cell.

Trial	Carrot	Green Bell Pepper	Beetroot	Number of Vegetables
22	Small cube	Stick	No	2
2	Stick	Stick	No	2
16	Small cube	Small cube	No	2
23	Small cube	Stick	Large cube	3
1	Stick	Stick	Stick	3
8	Large cube	Large cube	Large cube	3
15	Small cube	Small cube	Small cube	3

Hypothesis H3: Four additional trials were produced to allow pairwise comparison between samples differing only by the bell pepper color. The result is a third experimental design (Table 4) that allows measuring the impact of bell pepper color (yellow vs. green) for four different vegetable and shape combinations.

### 2.3. Vegetable Preparation

Each trial was prepared by weighing 300 g of vegetables. Trials with combinations of two vegetables were weighed at 150 g for each type and 100 g each for combinations of three vegetables, respectively. After preparation, the dishes were brought to the studio for photography (3840 × 2500 pixels). No background differences between the pictures and no edition were performed. In the picture, each vegetable dish was presented on a tray, with a spoon and a fork on two sides as references for the size of the veggies (Figure 1). In total, 27 pictures were taken with a set-up allowing to keep the aperture, ISO, and shutter speed constant across pictures.

### 2.4. Study Participants

As the samples were not manufactured ready-to-eat meals, seasonings, or culinary aids, we recruited online Nestlé employees across R&D centers and factories worldwide to reach enough respondents to draw reliable and robust conclusions from the statistical analysis. While it is acknowledged that the respondents may not be fully representative of the general population, it is important to note that we assumed their experience with raw vegetable preparation and consumption was similar to that of any consumers. Therefore, we did not anticipate any significant bias in the study. The study was assessed and approved internally as having met the ethical criteria to be considered as market research and was carried out in accordance with the Market Research Society’s (MRS) Code of Conduct.

Sample size 140 people participated in this study, which is above the 100 recommended number of participants for such quantitative consumer studies in adequation with external recommendations [22]. Each reported nationality, age, and gender using a completely anonymous online questionnaire. Eighteen participants did not fully complete the survey; hence, datasets of 122 participants (aged between 21 and 61, 72% women, 70% European)” were included in the statistical analysis.

### 2.5. Vegetable Evaluation

Participants were asked to score the perceived freshness of each vegetable preparation (based on a photo) using a 7-point scale anchored with “1 = Not fresh at all” and “7 = Very much fresh” (no definition of freshness was provided) following a full design with presentation plan of the pictures balancing the order and carry-over effects. A total of 122 responses per picture was collected.

### 2.6. Statistical Analysis

As a first step, a 2-way ANOVA was performed using “Vegetable” as a fixed factor and “Participant” as a random factor to measure the discrimination level between samples. Pairwise comparisons between samples were performed using the Fisher’s Least Significant Difference (LSD), calculated based on a 95% confidence level.

In a second step, the following data analysis plan was executed to address hypotheses H1a–c, H2, and H3 listed previously:Hypotheses H1a and H1b (21 trials, Table 2, result Section 3.1): a 4-way ANOVA was performed to quantify the impact of factors “Shape” (small cube, large cube, stick), “Carrot” (Yes, No), “Bell pepper” (Yes, No), and “Beetroot” (Yes, No) on the perceived freshness rating.Hypothesis H1c (21 trials, Table 2, result Section 3.1): a 2-way ANOVA was performed to quantify the impact of factors “Shape” (small cube, large cube, stick) and “Number of vegetables” (1, 2, 3) on the perceived freshness rating.Hypothesis H2 (7 trials, Table 3, result Section 3.2): the mean perceived freshness scores were compared using the global LSD calculated in the first step of the data analysis within group 1 combining two vegetables and within group 2 combining three vegetables as described below:○Group 1: Trials 2 (carrot and green bell pepper in stick), 16 (carrot and green bell pepper in small cube), and 22 (carrot in small cube and green bell pepper in stick).○Group 2: Trial 1 (carrot, green bell pepper, and beetroot in stick), 8 (carrot, green bell pepper, and beetroot in large cube), 15 (carrot, green bell pepper, and beetroot in small stick), and 23 (carrot in small cube, green bell pepper in stick, and beetroot in large cube) were compared using the global LSD calculated in the first step of the data analysis.Hypothesis H3 (4 trials, Table 4, result Section 3.3): A paired t-test was used to compare the mean perceived freshness score of each pair of samples differing only by the bell pepper color (samples 1 vs. 24, 13 vs. 25, 16 vs. 26, and 6 vs. 27) and assess the impact of “color” on perceived freshness.

All post-hoc comparison tests were performed using the Fisher’s LSD, and the threshold for statistical significance was set at 5% for all statistical analyses. The data analysis was conducted on IBM^®^ SPSS^®^ (28.0.1.1).

Results corresponding to hypotheses H1a–c, H2 and H3 are presented in result Section 3.1,Section 3.2, and Section 3.3, respectively.

## 3. Results

### 3.1. Impact of Shape, Presence/Absence of Vegetables, and Number of Vegetables on Perceived Freshness

Participants perceived very large differences in perceived freshness among the 27 samples (*F*(26,3146) = 36.9, *p* < 0.001); see ANOVA model in Appendix A, with 2 points difference between extremes on the 7-point freshness scale (Figure 2).

The shape significantly impacted perceived freshness (*F*(2/15) = 24.1, *p* < 0.001); see ANOVA model in Appendix A, with the stick shape significantly enhancing perceived freshness in comparison to the other shapes (Figure 3).

The presence of beetroot significantly decreased perceived freshness (*F*(1/15) = 14.3, *p* < 0.001), whereas the presence of carrot and green bell pepper did not significantly impact perceived freshness (*F*(1/15) = 3.0, *p* = 0.10, and (*F*(1/15) = 2.7, *p* = 0.12, respectively) (Figure 4) and see the ANOVA model in Appendix A.

The number of vegetables did not impact perceived freshness (*F*(2/12) = 0.23, *p* = 0.79) (Figure 5); see ANOVA model in Appendix A.

### 3.2. Impact of Shape Heterogeneity on Vegetable Perceived Freshness

Combining two vegetables (Figure 6a) and three vegetables (Figure 6b) in different shapes did not enhance perceived freshness in comparison to vegetables presented with the same shape, with the stick shape leading to the highest perceived freshness.

### 3.3. Impact of Bell Pepper Color on Vegetable Perceived Freshness

The green color of bell pepper significantly increased perceived freshness in comparison to the yellow color of bell pepper when vegetables were in sticks but not in large or small cubes (Figure 7).

## 4. Discussion

The aim of this study was to determine for the first time the impact of visual cues on perceived freshness perception of prepared vegetables. Indeed, existing research focuses on the perceived freshness of the whole vegetable, freshly picked and evaluated without any preparation. Stick shape was more associated with perceived freshness than small and large cubes, validating our first hypothesis (H1a), i.e., cubes are probably associated with processed vegetables commercialized frozen in bags or ambient in retorted glass or metal jars. In addition, vegetables in sticks may be easier to recognize compared with cube format and mainly to small cubes and thus perceived as less processed and fresher. Perceived freshness may still be associated with minimally processed vegetables, provided the physical integrity and color salience are preserved enough to recognize the vegetable [14]. Indeed, physical transformation reduces the naturalness of foods [23].

Validating hypothesis H1b, vegetable plates with beetroot were perceived as less fresh compared with plates without beetroot. The negative effect of beetroot on perceived freshness might be due to the dark color and association with bitterness; bitterness in vegetables is a negative driver of liking [24].

The number of ingredients in a plate, including vegetables, improves food consumer perception in different aspects, such as healthiness and nutrition, appealing to promote intake [16,20,21,24,25,26,27]. Contrasting colors in mixed salads promoted perceived freshness in a past study [28]. In the present study, a combination of vegetables did not enhance perceived freshness, invalidating the hypothesis H1c.

Combining two or three vegetables with different shapes reduced perceived freshness compared with the same vegetable combination in stick shape only, refuting hypothesis H2. Preparing and serving vegetables in different shapes could be considered an unusual and/or complex everyday culinary practice, reducing freshness perception.

We hypothesized the color modulates perceived freshness by replacing green bell pepper with yellow bell pepper (H3). We indeed observed that the green color was perceived to be fresher than the yellow color only for vegetables with a stick shape. It can be hypothesized that the green color enhanced perceived freshness only when the shape evokes a sufficient level of freshness through minimal physical transformation. A previous study showed that green color is strongly associated with vegetables in consumers’ minds [29].

## 5. Conclusions and Limitations

Visual cues modulated perception of freshness in vegetables. The results are particularly valuable for chefs and culinary arts professionals, catering professionals, and food industries involved in the preparation and/or manufacturing of prepared vegetables. Recommendations can also be applied at the consumer’s home and in school canteens via health care professionals and nutritionists.

By implementing these findings, they can enhance the freshness of their products and promote vegetable intake for a healthier diet. Indeed, shapes leading to a limited impact on the physical integrity of the vegetables should be promoted, as should combining vegetables with a similar shape preparation to probably evoke more authenticity and simplicity.

Green color association with vegetable freshness and the negative effect of beetroot on perceived freshness might be less generalized as likely associated to food eating habits and culture. In the present study, as 70% of participants were European, the role of culture could not be explored, and dedicated cross-cultural research is then required.

No freshness definition and no contextual background were proposed to the participants not to influence them, as freshness is a multifactor perception. Although visual drivers of vegetable freshness perception may vary, for instance, between vegetables expected as raw in a cold salad or as cooked in hot dishes, observed significant differences demonstrate that freshness shared a common, even though partly, understanding among participants.

Finally, the respective contribution of vegetable flavor and textural cues to perceived freshness, in addition to appearance, also needs to be taken into consideration in future studies.

## Figures and Tables

**Figure 1 foods-13-03342-f001:**
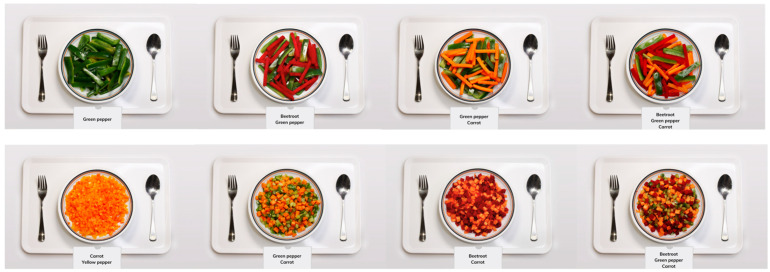
Examples of vegetable dish pictures.

**Figure 2 foods-13-03342-f002:**
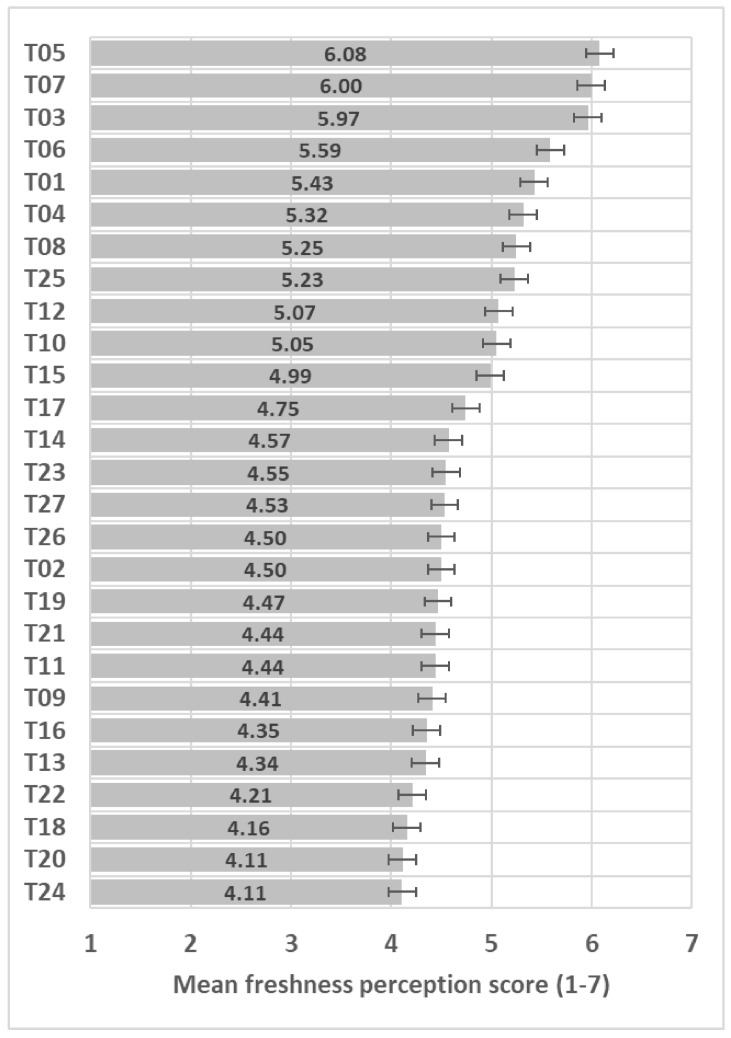
Freshness mean score (+/−LSD/2) of the 27 samples included in the study with LSD = 0.27. When the LSD/2 crosses over, the trials do not show significant differences.

**Figure 3 foods-13-03342-f003:**
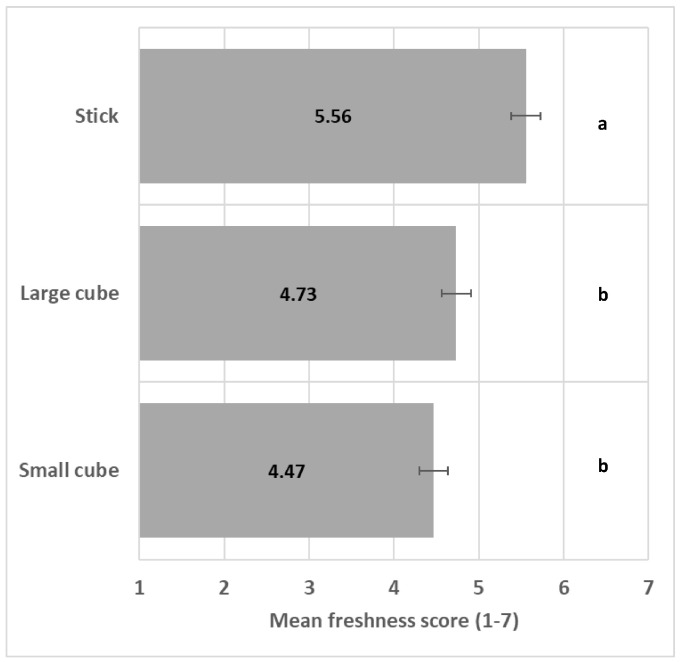
Impact of shape on freshness mean score (+/−LSD/2) with LSD for Shape = 0.35. Levels with the same letter are not significantly different.

**Figure 4 foods-13-03342-f004:**
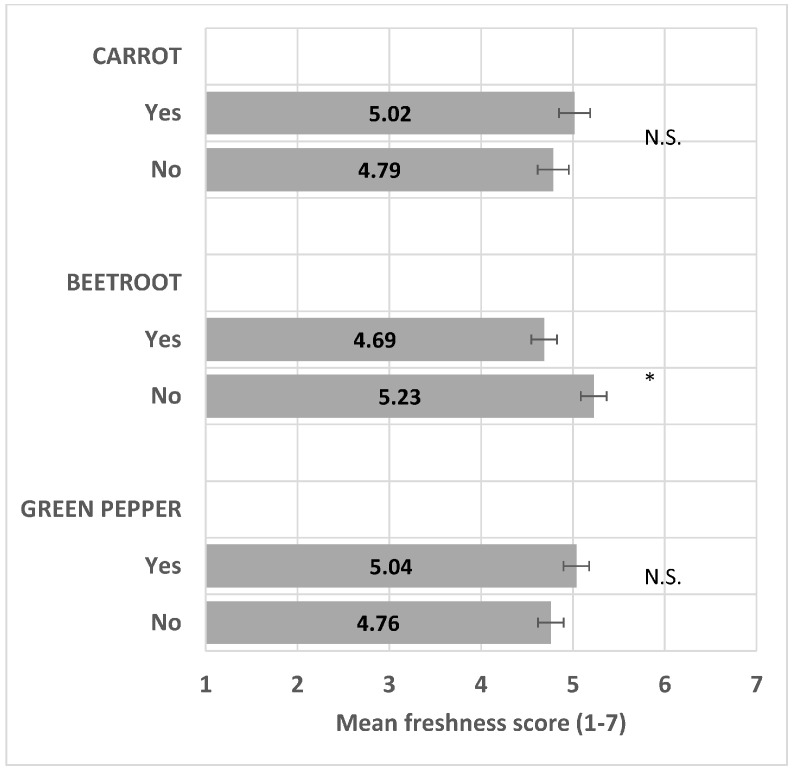
Impact of vegetable presence/absence (Yes/No) on freshness mean score (+/−LSD/2) with LSD = 0.28. “*” means levels within the pair are significantly different. “N.S.” means levels within the pair are not significantly different.

**Figure 5 foods-13-03342-f005:**
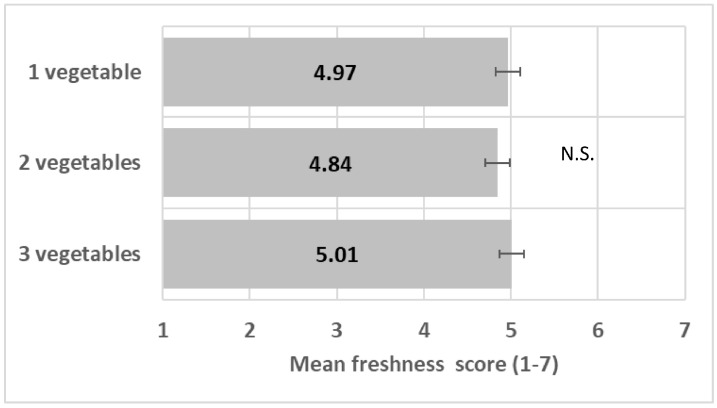
Impact of number of vegetables on freshness mean score (+/−LSD/2). “N.S.” means levels are not significantly different.

**Figure 6 foods-13-03342-f006:**
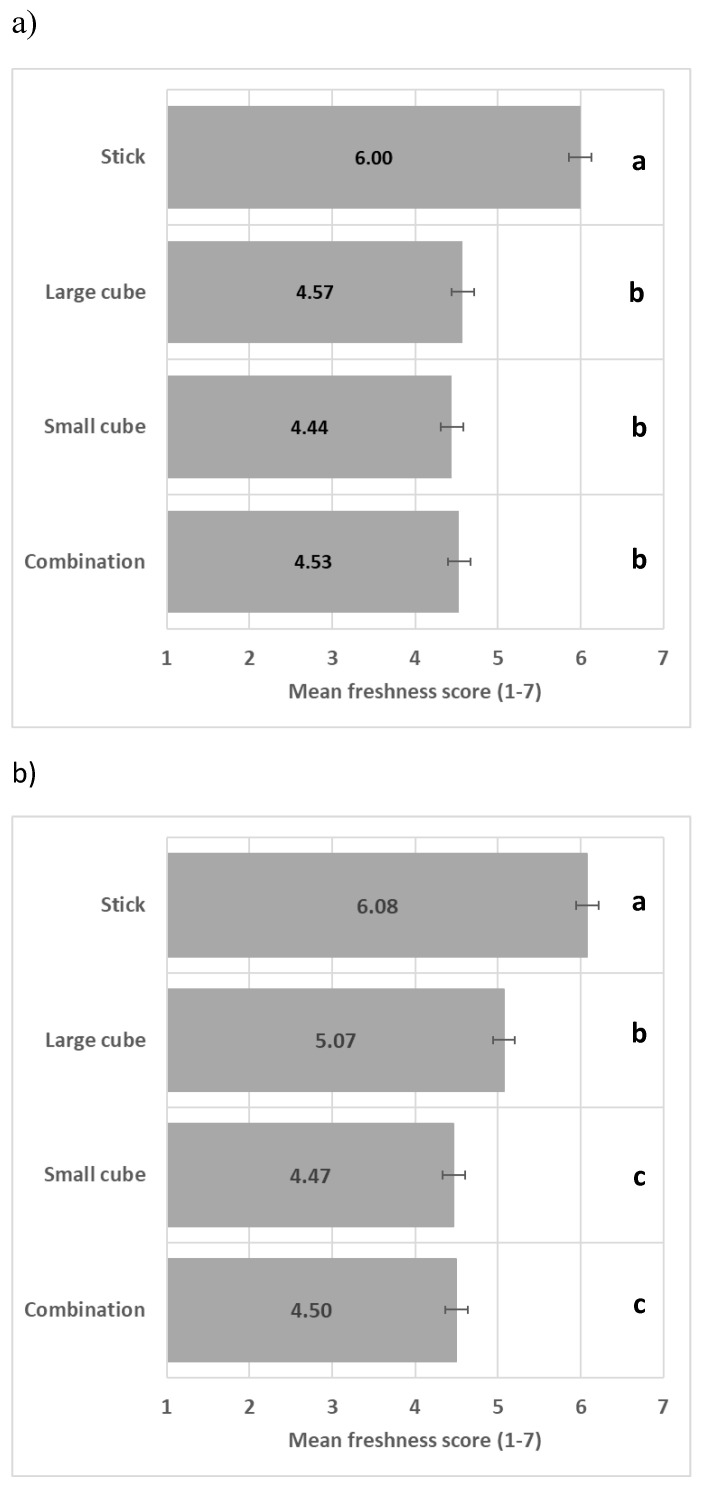
Impact of vegetable shape heterogeneity on perceived freshness mean score (+/−LSD/2) when carrot and green bell pepper (**a**) and carrot, green bell pepper, and beetroot (**b**) are present (LSD = 0.27). Levels with different letters are significantly different in perceived freshness.

**Figure 7 foods-13-03342-f007:**
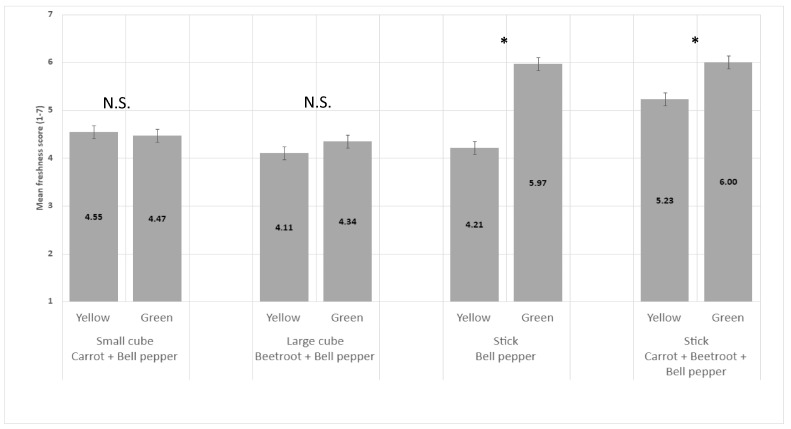
Impact of bell pepper color on perceived freshness mean score (+/−LSD/2) of vegetables in stick and cube shapes. “*” means difference in perceived freshness between the two levels within each pair is significant. “N.S.” means levels within the pair are not significantly different.

**Table 1 foods-13-03342-t001:** Vegetable shapes.

Shape	Size (mm)
Stick	8 × 8 × 50
Large cube	10 × 10 × 10
Small cube	5 × 5 × 5

**Table 4 foods-13-03342-t004:** Experimental design for color evaluation. Samples 24, 25, 26, and 27 in white cell are similar to samples 1, 13, 16, and 6 in gray cell, respectively, except that green pepper was replaced by yellow pepper.

Trial	Shape	Carrot	Bell Pepper	Beetroot	Number of Vegetables
24	Stick	Yes	Yes—Yellow	Yes	3
1	Stick	Yes	Yes—Green	Yes	3
25	Large cube	No	Yes—Yellow	Yes	2
13	Large cube	No	Yes—Green	No	1
26	Small cube	Yes	Yes—Yellow	No	2
16	Small cube	Yes	Yes—Green	No	2
27	Stick	No	Yes—Yellow	No	1
6	Stick	No	Yes—Green	No	1

## Data Availability

The original contributions presented in the study are included in the article/Appendix A, further inquiries can be directed to the corresponding author.

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
