# Peer review of "Impact of Visual Cues on Consumers’ Freshness Perception of Prepared Vegetables"

_foods, 2024, doi:10.3390/foods13203342_

Round 1

Reviewer 1 Report

Comments and Suggestions for Authors

Although I find the study somewhat interesting I'm not sure the sample size, the design of the experiment, or the conclusion provide adequate testing or any true explanation as to why this study is of value.

Only 122 respondents for 27 different images? That is less than 5 respondents per image - also there is no reference provided support for the statement that 140 responses was 'above the internally recommended number of participants'?! Internally recommended where? What is the actual cut off you went with? is 4.5 responses per a minimum number recommended somewhere?

Relatedly, why only use R&D employees from Nestle? Why not ask a general consumer group rather than a group of individuals who work for a food manufacturer?

The authors need to provide the full ANOVA reports as well, though I feel like the sheer number of trials that were run could have been trimmed or broken up more directly so that it is easier for the reader to follow and understand what each trial is investigating.

I also have an issue with Page 2 Lines 58-60 - this reference is nearly a decade old so not really 'recent years' but it is also terribly sexist. I would also venture an educated guess that there is research suggesting that in 'recent years' there has been an uptick in vegetarian diets which would be a counterpoint to the argument made here. I feel this sentence can be removed or at least revised.

The conclusion is very weak and provides little to no explanation as to what we should do with the information gleaned from this study, or how it is any different than past studies. Who benefits from this study and how?

Author Response

Dear Reviewer,

Replies to your comments have been addressed in the attached word documents.

With best regards,

David Labbe

Reviewer 2 Report

Comments and Suggestions for Authors

This study focused on the role of visual cues of common vegetables (carrot, beetroot, bell pepper) on perception of freshness. The manuscript lacks novelty and the results are expected.

INTRODUCTION

- The innovation of the manuscript is not clear stated.

MATERIAL AND METHODS

- The approval of the ethics committee should be included and, as soon as it is an academic paper, it should not be an internal approval. 

-  "We recruited on-line Nestlé employees across R&D centers and factories worldwide.  We considered them as representative consumers who frequently prepared and con sumed raw vegetables at home or experienced them in restaurants.". This is not right. Why would they be a representation of the consumers in a general view. The authors should include more information. 

Author Response

Dear Reviewer,

Replies to your comments are added in the attached Word document.

We hope we have addressed them in a satisfactory manner.

With best regards,

David Labbe

Reviewer 3 Report

Comments and Suggestions for Authors

Dear authors,
Thank you for submitting your research. Freshness is a topic that is highly discussed in the field, with different points of view and interpretations of raw or processed food. Although I've appreciated your research, I'd like to give you some suggestions to improve it.

Here are some notes for you:

- At the end of the Introduction, the authors should declare how the paper is divided to make the reader acknowledge what awaits.

The description of your methodology seems a bit rushed. I suggest you improve it by giving more context and explaining the reason behind your choices.

- The result section lacks in-depth considerations. The authors should provide more information and match it with theoretical explanations.

- Same considerations for the discussion section. Providing more details, explanations, and projected implications about the acceptance or not of your hypotheses may enrich your contribution to the field and make it worth reading and citing.

- As previously stressed, your paper lacks the "actual contribution". It doesn't mean your results are useless, but you didn't value them appropriately. In the "Conclusions and Limitations" section, you should stress your actual contribution, underlying the usefulness of your results and the implications deriving from them, especially for food businesses. As for the definition of freshness, giving your respondents a proper idea of what you mean (differentiating between the different shades of freshness) may benefit your study.

I hope you'll find my comments helpful.

Good luck with your research!

Author Response

(The authors gave the same response as above.)

Round 2

Reviewer 1 Report

Comments and Suggestions for Authors

The edits made make the paper far easier to read/follow, I appreciate the work done by the authors. I do think adding the tables that are in the response document as additional materials at the end of the manuscript would be a good idea.

Author Response

Dear Reviewers,

The ANOVA tables are now added in a Supplemetary Data file and are referenced in the result section.

Sincerely,

David Labbe

Reviewer 2 Report

Comments and Suggestions for Authors

The authors performed the suggested changes. 

Author Response

The authors have effectively addressed the reviewer comments.